# *Burkholderia cepacia* complex (Bcc) in goats: First report in Bangladesh

**Abdullah Al Mamun**[1], **S. M. Sujan Ashraf**[2], **Amir Hamza Shuvo**[1], **Farhan Ibne Siddique**[3], **Hasan Khan**[4], **Naeem Ahammed Ibrahim Fahim**[5], **Chandra Shaker Chouhan**[1], **Farzana Yeasmin**[1], **Md. Mahbub Alam**[1], **Md Tanvir Rahman**[5], **Md. Siddiqur Rahman**[1]*

1 Department of Medicine, Faculty of Veterinary Science, Bangladesh Agricultural University, Mymensingh, Bangladesh, 2 Department of Pathology, Faculty of Veterinary Science, Bangladesh Agricultural University, Mymensingh, Bangladesh, 3 Armed Forces Medical College, Dhaka Cantonment, Dhaka, Bangladesh, 4 Department of Entomology, Faculty of Agriculture, Bangladesh Agricultural University, Mymensingh, Bangladesh, 5 Department of Microbiology and Hygiene, Faculty of Veterinary Science, Bangladesh Agricultural University, Mymensingh, Bangladesh

* siddique.medicine@bau.edu.bd

## Abstract

The *Burkholderia cepacia* complex (Bcc) comprises a diverse group of opportunistic pathogens that are critically relevant to human and animal health. Bcc induces life-threatening infections of the lung in human patients with cystic fibrosis and chronic granulomatous disease and causes mastitis and abscesses in sheep and goats. Therefore, this research aimed to identify the presence of *Burkholderia cepacia* complex (Bcc) in goats in Bangladesh using serological tests such as the Glanders Rapid Detection Test Kit (GRDTK) and Enzyme-linked immunosorbent assay (ELISA) and molecular tests such as PCR, Sanger sequencing, and phylogenetic analyses. A total of 40 goat blood samples were collected, comprising 30 samples from goats exhibiting a history of abortion accompanied by respiratory symptoms and 10 samples from clinically healthy control goats. Among the 30 samples from symptomatic goats, eight isolates (26.67%) were observed to grow in Luria-Bertani broth, while there was no growth from the control group. Five (62.50%) were found positive for bacterial presence when analyzed using the 16S rRNA gene-specific PCR assay for broth-positive samples. In the Genomix GRDTK and ELISA tests, six isolates were identified as positive for *Burkholderia* spp. Of the eight broth culture-positive samples, two (25%) were found to be positive by genus-specific PCR using *groEL* primers and species-specific amplification using *zmpA* primers. Phylogenetic analysis of positive goat samples confirmed the presence of the Bcc, which showed 100% similarity with the strains from India, Japan, and China. The discovery is the first detection of Bcc in animals in Bangladesh. It raises significant concerns for animal and public health. These findings highlight the critical need for strengthened diagnostic strategies, improved microbiological surveillance, and further research into Bcc's

**Data availability statement:** All relevant data are within the manuscript and its supporting information files.

**Funding:** This research was supported by the University Grants Commission of Bangladesh (UGC) under the project titled 'Prevalence and Risk Factors of Glanders in Humans and Horses' (Project Code: 2024/17/UGC to SR) and the Ministry of Education (MoE) under the project titled 'Development of diagnosis and control strategies against brucellosis in small ruminants and humans' (Project Code: 2022/16/MoE to SR). The funders had no role in study design, data collection and analysis, the decision to publish, or the preparation of the manuscript.

**Competing interests:** The authors have declared that no competing interests exist.

epidemiology and zoonotic potential within the One Health framework to understand better and mitigate its zoonotic risks.

## Introduction

The genus *Burkholderia* constitutes a significant clade within the class β-proteobacteria. It consists of over 70 species that survive in various environments, including soil, water, and host organisms [1]. Among *Burkholderia* species, one notable subgroup is the *Burkholderia cepacia* complex (Bcc). It is a collection of 24 genetically related species famous for their diversity and ecological adaptability [2,3]. *B. cepacia* itself, formerly known as *Pseudomonas cepacia*, was first identified in 1950 by Burkholder as the causal agent of bacterial root disease in onion bulbs [4,5]. It is a Gram-negative, non-fermenting, non-spore-forming, oxidase-positive aerobic bacillus, ubiquitous in soil, water, and on plant surfaces [6–9].

Bcc species are increasingly recognised as emerging pathogens, especially within healthcare facilities. Infections represent significant life-threatening risks for individuals with cystic fibrosis (CF) and chronic granulomatous disease [7,8]. *B. cenocepacia*, a highly virulent species within the Bcc group, is linked to a mortality rate that is up to five times higher than that of other species [10–12]. Infections in healthy individuals are rare but have been reported in cases linked to contaminated medical products such as ultrasound gels, chlorhexidine solutions, water for injection, anaesthetic eye drops, liquid docusate, and mouthwash solutions [12–14].

Invasive Bcc infections are reported globally, presenting a considerable challenge due to their antibiotic resistance as well as their ability to form biofilms [15]. Clinically, infections manifest as respiratory distress, bacteremia, wound infections, and systemic sepsis, particularly in CF patients, where they contribute to progressive lung deterioration and a high rate of mortality [6,10]. Cases of deep neck abscesses, pyomyositis, recurrent endophthalmitis, and infective spondylitis have also been documented [16].

Though Bcc is generally responsible for human infections, it has been progressively reported in animals, which highlights its potential for zoonotic transmission. Bcc infections have been well documented in horses, cats, dogs, piglets, birds, and sheep, often presenting as respiratory diseases, abscesses, cellulitis, and systemic infections [1,10]. Recently, Bcc has been identified in several cases, like purulent cellulitis with subcutaneous abscesses in cats [17], recurrent balanitis in horses [18], and deep pyoderma in dogs [8]. Moreover, multisystemic fibrin thromboses in felines indicated septicemia, identical to "cepacia syndrome" reported in humans [8,16,18–20]. Pathological alterations identified in parrots encompassed subcutaneous oedema, hyperaemia, hemorrhages, and necrosis in various organs, indicative of *Burkholderia* spp. infections [4]. In poultry, *Burkholderia* spp. are associated with systemic or localised diseases such as sinusitis, keratitis, septicaemia, and neurological signs [4]. There are also reports that Bcc causes mastitis in sheep [21] and lung, liver, kidney, and spleen abscesses in goats [22]. Due to its capacity to infect both

humans and animals, experimental infections in mice indicate that Bcc may facilitate cross-species transmission, necessitating further research [10].

As *B. cepacia* has been recorded in human clinical cases in Bangladesh [23], it raises concerns about being spread in the environment through contaminated hospital equipment, which can cause infections in livestock animals. However, their occurrence in livestock and veterinary contexts has yet to be investigated. Therefore, this study aimed to detect the presence of Bcc in goats in Bangladesh, marking the first attempt to investigate its potential role in animal infections in that region. Understanding Bcc's prevalence in livestock will provide valuable insights into its epidemiology, transmission dynamics, and zoonotic risks in Bangladesh.

## Materials and methods

### Ethical approval

All animal-related procedures and methods were carried out following the guidelines by the Animal Welfare and Experimentation Ethical Committee of the Bangladesh Agricultural University, Mymensingh (Ethical approval number - AWEEC/BAU/2023(55))

### Study design and sample collection

A cross-sectional investigation was performed involving 40 Black Bengal goats, including 10 clinically healthy control goats and 30 goats presenting with a history of abortion and respiratory signs. All goats were sampled from the same farm environment, and control goats showed no clinical signs of respiratory or systemic illness. Blood samples were collected from the Goat, Sheep, and Horse Farm (24.72923 N, 90.42207 E), Bangladesh Agricultural University (BAU), Mymensingh district (24.7460° N, 90.4179° E) from May 2023 to October 2024 (Fig 1). Approximately 8–10 mL of blood was collected from each goat & kept in clot activator tubes for 30 minutes at room temperature for clotting and serum separation. Then the serum was immediately transferred to the Department of Medicine lab (24.72573 E, 90.43690 N) at Bangladesh Agricultural University, Mymensingh, using an icebox and stored at 4°C overnight. Thereafter, the sera were transferred to labelled test tubes and centrifuged at 2500 rpm for 10 minutes. Clear sera were then transferred to vials and stored at −20 °C until further use.

### Bacterial culture and DNA extraction

100 microlitres of serum were added to 5 mL of Luria-Bertani broth, Miller (HiMedia, Maharashtra, India), and incubated aerobically at 37°C overnight to culture the bacteria. Then, according to [24], the DNA was extracted from bacterial colonies using the boiling method. To put it briefly, a 1 mL sample of the enriched culture was first centrifuged for 5 minutes at 5000 rpm. After discarding the resultant supernatant, 200 µL of phosphate buffer solution was added to create a suspension. After boiling and cooling the suspension for ten minutes, the suspension was centrifuged for ten minutes at 10,000 rpm. For future study, the resultant supernatant, which contained the genomic DNA, was gathered and kept at −20°C.

### Polymerase chain reaction (PCR)

The PCR protocol and primers, as shown in Table 1, were used in this study to identify unknown bacteria according to [25]. To create the reaction mixture, 2.5 µL of 10x Taq buffer, 1 µL of 50 mM $MgCl_2$, 2.5 µL of 2 mM dNTPs, 0.2 µL of Platinum® Taq Polymerase (5 U µL$^{-1}$, Invitrogen™), 5 pmoles of Primer 27F, 10 pmols of Primer 1492R, and 8 ng of DNA template were combined with ultrapure water until the final volume was 25 µL. The PCR amplification starts with DNA denaturation at 96°C for 4 min, followed by 30 cycles of denaturation at 94°C for 30 seconds, annealing at 57°C for 30 seconds, and elongation at 72°C for 1 min. The final extension is done at 72°C for 10 min. PCR products were analyzed by 1% agarose gel electrophoresis at 120 V for 40 minutes and compared with a DNA ladder.

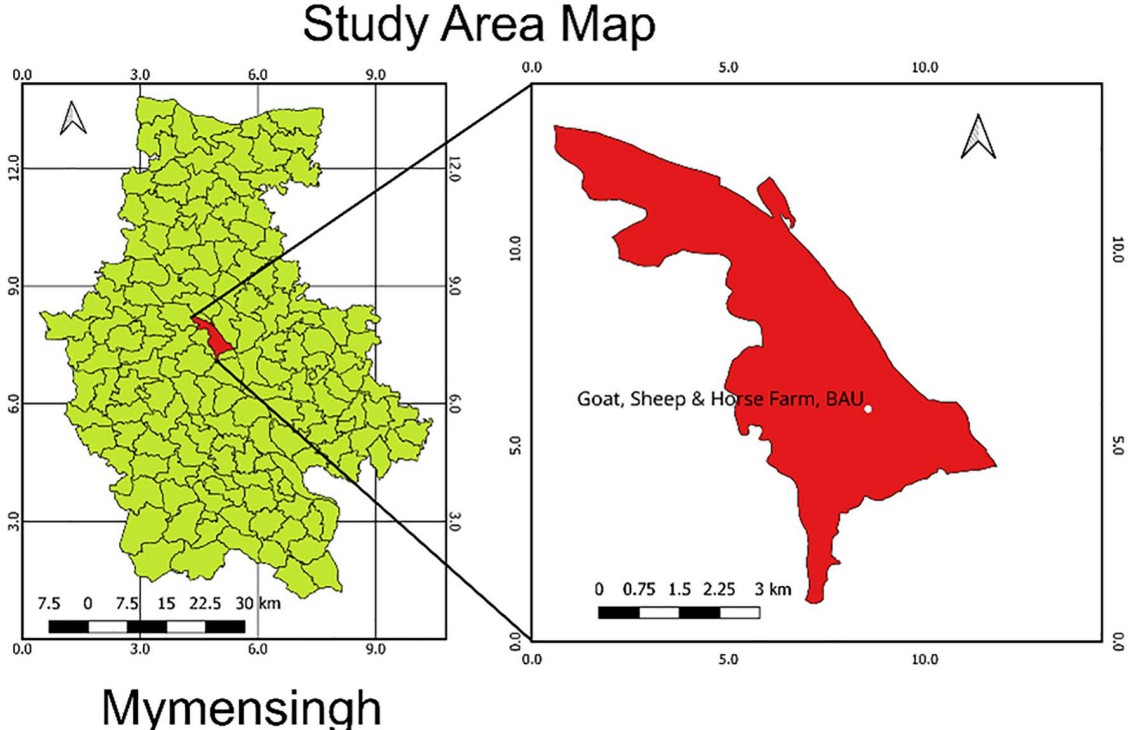

**Fig 1. Study Area Map (Mymensingh District). The study area map was created using QGIS Desktop 3.40.0.**

**Table 1. Primers used in this study.**

| Targeted factors | Target Region | Primer Sequence (5´-3´) | Annealing Tm (°C) | Amplicon Size | References |
|---|---|---|---|---|---|
| Unknown Bacteria | 16S rRNA gene | 27F: AGAGTTTGATCCTGGCTCAG | 57 | 1400 bp | [25] |
| | | 1492R: TACGGYTACCTTGTTACGACTT | | | |
| *Burkholderia* | *groEL* gene | F: CTGGAAGACATCGCGATC | 52 | 139 bp | [26] |
| | | R: CGTCGATGATCGTCGTGTT | | | |
| *Burkholderia cepacia* | *zmpA* Gene | F: ACCCTCGCGAGTCTCAG | 52 | 147 bp | |
| | | R: TTGTGGCCCGGCGAACTT | | | |

## Molecular identification by sanger sequencing and phylogenetic analysis

A single representative amplicon was chosen at random for partial sequencing. PCR products underwent purification, followed by the generation of single-stranded products through cycle sequencing PCR utilising either forward or reverse primers. PCR products were analyzed on a Sanger sequencer employing the dideoxy chain termination technique at Wuhan Tianyi Huayu Gene Technology Co., Ltd [27]. The acquired sequences were meticulously examined for possible sequencing errors using MEGA 11 [28]. They were subsequently employed to query the GenBank online database nucleotide collection (nr/nt) utilising the megablast algorithm, which is optimised for highly similar sequences, within the basic local alignment search tool (BLAST) online application (https://blast.ncbi.nlm.nih.gov/Blast.cgi). The phylogenetic tree was generated using 40 partial 16S rRNA gene sequences of *Burkholderia*, of which 5 were sequenced in this study. The

neighbor-joining method was employed for phylogenetic tree construction, and evolutionary distances between sequences were determined using the maximum composite likelihood method.

To provide statistical support for the tree's branches, bootstrap analysis with 1000 replications was performed. The analyses were conducted and visualised with MEGA11 software [28]. The phylogenetic tree illustrates the sequences from this study, marked in red, denoting their position and relationship with other *Burkholderia* species sequences. Bootstrap support values are indicated at the nodes of the tree, providing statistical confidence for the inferred relationships.

### Serological tests for the *Burkholderia*

To validate the presence of the *B. cepacia* complex in the original serum samples, the Glanders Rapid Detection Test Kit (Genomix Biotech, USA) and Enzyme-linked immunosorbent assay (ELISA) were performed. The Glanders Ab Rapid Detection Test Kit was carried out according to the manufacturer's instructions in the supplied catalogue. 10 µL of the serum sample was added to the sample well on the test cassette using the dropper. Two drops of test buffer were placed in the identical sample well on the test cassette. After 20 minutes, a line appeared in both the control (C) line and the test (T) line, indicating a positive sample. Still, a line appearing just in the control (C) line region was classified as a negative sample. In enzyme-linked immunosorbent assay, *Burkholderia* antibodies in the serum were detected for further confirmation using the Glanders Antibody Detection ELISA Test Kit (Genomix Biotech, USA).

In brief, according to [29], the required number of wells was placed in the frame, and a protocol sheet was prepared. Then, 100µl of control sera (positive/negative) were added to the wells in duplicates, and a 1:100 dilution was ready for each test sample by mixing 200 µl of sample diluent with 2 µl of serum; 100 µl of each diluted sample was transferred to the respective wells. The plate was covered and incubated at 37°C for 1 hour. After incubation, the wells were washed four times with 300 µl of diluted wash buffer, residual liquid was removed by tapping, and 100 µl of conjugate solution was added to each well. The plate was incubated again at 37°C for 1 hour. Following this, the plate was washed five times, 100 µl of chromogen solution was added to each well, and it was covered with foil and incubated at 37°C for 10–15 minutes in the dark until color developed. The reaction was stopped by adding 100 µl of stop solution, and OD was measured at 450nm using a Multiskan TM FC Microplate ELISA reader (Thermo Fisher Scientific, USA). The positivity percentage (PP%) was calculated to identify the *Burkholderia* genus from the reference sample as per the following formula:

Percent positivity (PP%) = [(OD450 Sample serum - OD450 Negative control) / (OD450 Positive control - OD450 Negative control)] * 100.

The result was matched with PP value (< 20% = negative, 20–25% = equivocal, > 25% = positive) for ELISA for **Burkholderia spp.**

### PCR detection of genus and species

The DNA was amplified using a *Burkholderia* genus-specific primer (*groEl*) and *B. cepacia* species-specific primer (*zmpA*), as given in **Table 1**. A final volume of 50 µL was used for the PCR reactions, which included 1 PCR buffer, 2.5 mM $MgCl_2$, 0.2 mM of each dNTP, 0.2 mM of each primer, 1.25 U of Taq DNA polymerase, 5 µL of DNA template, and distilled water. According to [26], amplification consisted of 35 cycles of denaturation at 94°C for 30s, annealing at 52°C for 30s, and extension at 72°C for 45s. The initial denaturation was carried out at 94°C for 5 min. After the last cycle, a previous extension lasting an additional two minutes was performed at 72°C. A 100 bp DNA ladder (Invitrogen™) was used to compare the band on an agarose gel. The amplified PCR products were confirmed on agarose gels (1%) stained with ethidium bromide on the gel documentation system (EZEE Clearview UV transilluminator).

### Statistical analysis

The data was analyzed using an Excel 2013 spreadsheet (Microsoft Office 2013, Microsoft, Los Angeles, CA, USA). Moreover, binomial 95% confidence intervals were generated in GraphPad Prism version 8.4.3 (GraphPad Software, Inc.).

## Results

### Bacterial isolation and molecular identification

Bacterial growth was observed in eight (26.67%, 95% CI: 14.18–44.45) out of 30 symptomatic goats on Luria-Bertani broth inoculated with serum samples (S1 **Fig**), whereas no growth was detected in samples from healthy control goats. In PCR for detecting unknown bacteria from broth culture-positive isolates with universal bacterial primers, five (62.50%, 95% CI: 30.57–86.32) out of eight tested positive for bacterial presence when analyzed using the 16S rRNA PCR assay (Fig 2).

### Molecular characterisation and phylogenetic analysis

Through BLASTn analysis, a complete alignment of 100% homology was observed between our nucleotide sequences of the five selected amplicons and previously documented *Burkholderia cepacia* complex sequences, confirming the species identification of our isolates. The sequences have been deposited in GenBank under the following accession numbers: PQ657137.1 (*Burkholderia contaminans* BAU-MSR-AAM-01), PQ657138.1 (*B. ambifaria* BAU-MSR-AAM-02), PQ657139.1 (*B. contaminans* BAU-MSR-AAM-03), PQ657140.1 (*B. paraquae* BAU-MSR-AAM-04), and PQ764482.1 (*B. cepacia* BAU-MSR-AAM-05). In phylogenetic analysis, the sequenced *Burkholderia* isolates from Bangladesh (BAU-MSR-AAM-01, PQ657137; BAU-MSR-AAM-02, PQ657138; BAU-MSR-AAM-03, PQ657139; BAU-MSR-AAM-04, PQ657140; BAU-MSR-AAM-05, PQ764482) exhibited strong genetic relationships with *Burkholderia* strains from various global sources. The bootstrap values at the nodes ranged from 37% to 100%, indicating varying degrees of confidence in the tree's branching. The red-marked samples were closely related to strains such as **KT906686.1** (*Burkholderia cepacia* strain PSTJ15) from India (bootstrap value: 100%), **KT719943.1** (*Burkholderia contaminans* strain BAU-MSR-AAM 05) from Bangladesh (bootstrap value: 58%), and **MH298779.1** (*Burkholderia contaminans* strain CZ-1) (bootstrap value: 38%). These results confirm the genetic similarity of the isolates with closely related *Burkholderia* species, contributing valuable insights into their phylogenetic diversity (Fig 3).

### Serological confirmation of *Burkholderia*

Six (75%, 95% CI: 40.93–92.85) out of eight goat samples tested positive for the GARDTK (S2 Fig). The same result was also observed in ELISA, as six (75%, 95% CI: 40.93–92.85) out of eight goat samples tested positive (S3 Fig), as shown in Table 2.

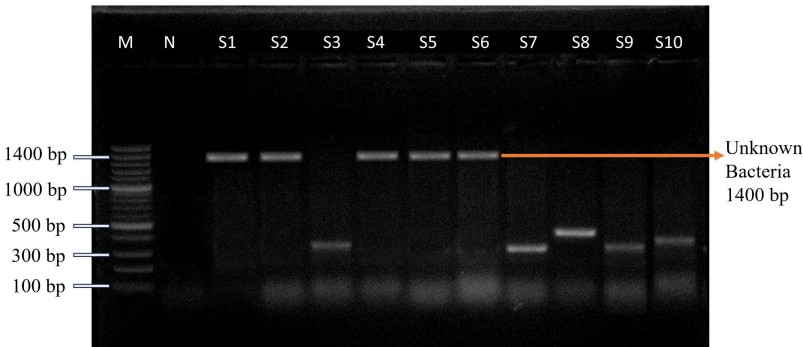

**Fig 2. Gel electrophoresis image of amplicons produced from the 16S rRNA PCR assay (1400 bp).** Lane M for 100 bp ladder (Invitrogen™); Lane N for Negative control and lanes S1-S10 are for samples. Lane S1-S6 for broth-positive samples of suspected animals & lane S7-S10 for samples of the control group. Here, S1, S2 and S4-S6 are positive lanes (1400 bp), and S3, S7-S10 are negative lanes for unknown bacteria.

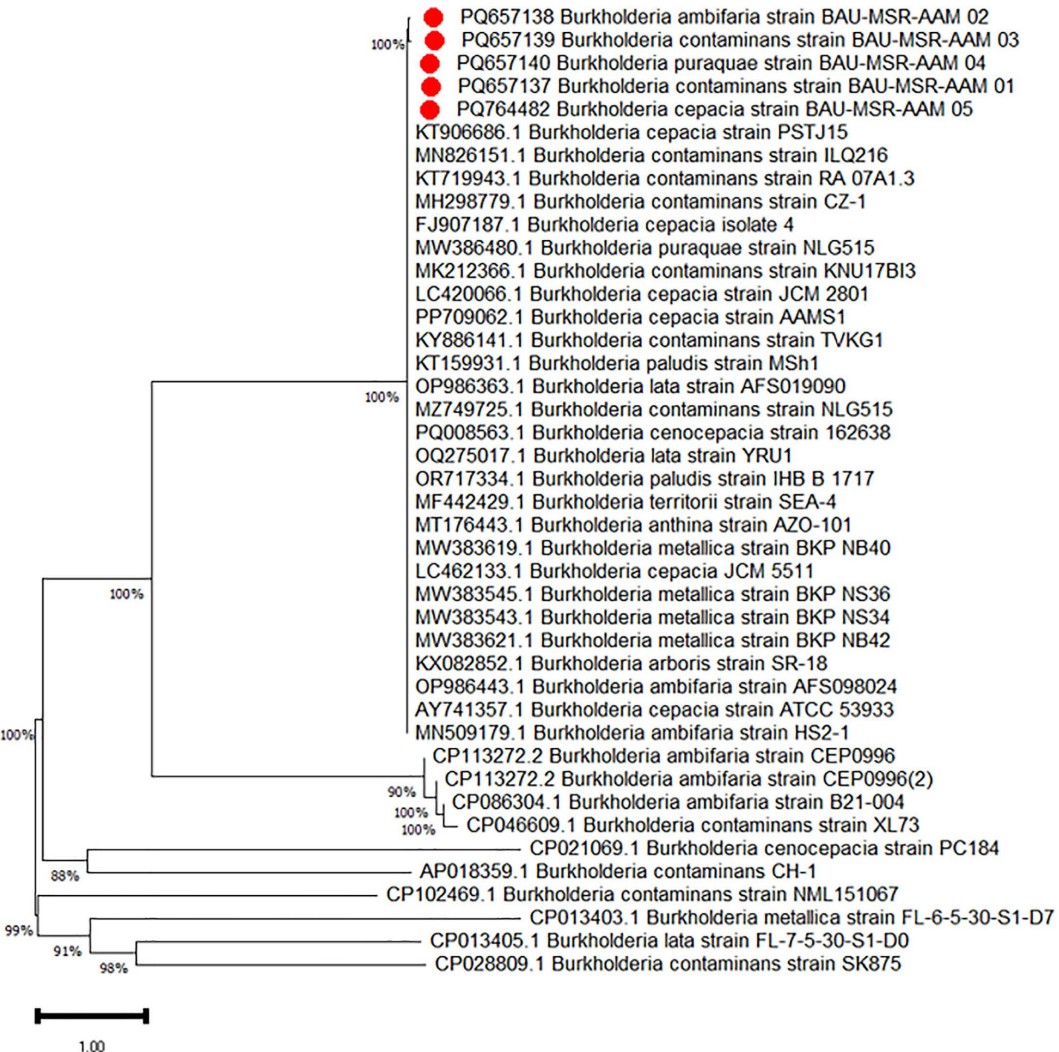

**Fig 3. The phylogenetic tree was generated using 40 partial 16S rRNA gene sequences of *Burkholderia*, of which 5 were sequenced in this study.** The neighbor-joining method was employed for phylogenetic tree construction, and evolutionary distances between sequences were determined using the maximum composite likelihood method. MEGA11 analyzed and visualised the phylogenetic tree. This study's sequences are marked in red on the phylogenetic tree to show their relationship to other *Burkholderia* species' sequences.

**Table 2. Prevalence of *Burkholderia cepacia* complex in Rapid Detection Test Kit, ELISA, PCR.**

| Test name | Category (N) | n (%, 95% CI) |
|---|---|---|
| **Glanders Antibody Rapid Detection Test Kit** | **Genus detection [8]** | 6 (75%, 95% CI: 40.93–92.85) |
| **ELISA** | | six (75%, 95% CI: 40.93–92.85) |
| **PCR** | **Genus detection [8]** | 2 (25%, 95% CI: 7.14–59.07) |
| | **Species detection** | 2 (25%, 95% CI: 7.14–59.07) |

**Here, N = total number of samples tested for different tests, n = positive sample for each test, and CI = confidence interval.**

## PCR detection of *Burkholderia* Genus and species

The genus-specific PCR with *groEL* primers resulted in two (25%, 95% CI: 7.14–59.07) positive samples out of eight broth culture-positive samples (Fig 4). Amplification with *zmpA* primers for species-specific production produced a distinct ~147 bp band in two (25%, 95% CI: 7.14–59.07) out of eight broth culture-positive samples, confirming the presence of the *B. cepacia* complex in the samples (Fig 4), as shown in Table 2.

## Discussion

This is the first report of *B. cepacia* complex (Bcc) in goats in Bangladesh. As for Bcc's potential effects on animal health and public health, our results will provide crucial information about its presence in the goat blood, which has the potential to be transmitted to the environment. However, environmental or human sampling was not included in this study; therefore, zoonotic transmission remains hypothetical. Additionally, the identification of Bcc in goats indicates that small ruminants could act as an unnoticed reservoir for this opportunistic pathogen, necessitating further research into its epidemiology, pathogenicity, and zoonotic potential. Given the diverse environmental niches Bcc occupies, its presence in livestock highlights possible transmission routes that could affect both animal and human populations.

In our study, we observed respiratory symptoms in *Bcc*-infected goats, consistent with findings from other studies [28–31]. Before this study, one additional doe in the similar herd (total number of population was 41 animals) had identical respiratory problems & cellulitis, resulting in death without any clinical signs. However, a post-mortem was not conducted since the farmer had not sought veterinary guidance. Therefore, the reason for the problem was never examined. In PCR for unknown bacteria, we found 62.5% for bacterial presence when analyzed using the 16S rRNA PCR assay. However, 75% of isolates were found positive for the *Burkholderia* genus in serological tests such as GARDTK and ELISA. The discrepancy observed among culture, PCR, and serological methods may be attributed to differences in sensitivity and specificity. Bacterial culture followed by molecular confirmation (PCR and sequencing) was considered the gold standard for definitive identification in this study. There is a report that *the Burkholderia genus is attributed to causing* chronic diseases in goats [32]. Although goat research is lacking, pure *B. cepacia* cultures were identified from 1 of 33 (3.0%) CMT-negative ewes and 64 of 96 (66.7%) CMT-positive ewes [21]. Our selected farm for this study evidenced endemic mastitis; however, it was uncertain whether the doe had subclinical mastitis, as B. cepacia infection was previously associated with outbreaks of ovine subclinical mastitis [21]. B. cepacia is regarded as an environmental pathogen, indicating that the doe may have acquired the infection from an environmental source. Furthermore, there was no record of newly imported animals, and this farm did not implement dry-period antibiotic infusion. Antibiotics administered during the dry period have been documented to induce outbreaks of Pseudomonas aeruginosa mastitis in cattle because of contamination [33]. The

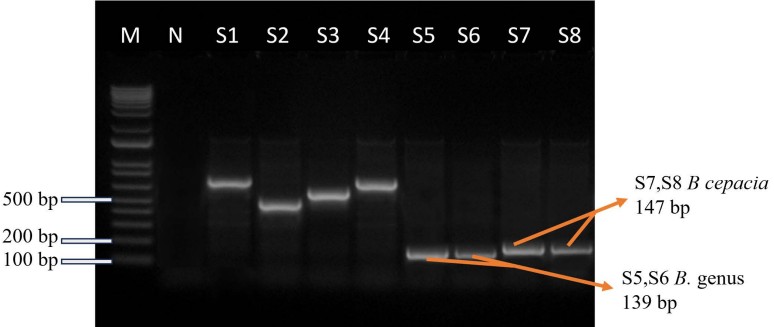

**Fig 4. Gel electrophoresis image of amplicons produced from the groEL gene PCR assay (139 bp) for the *Burkholderia* genus and the zmpA gene PCR assay (147 bp) for *Burkholderia cepacia*.** Lane M for 100 bp ladder (Invitrogen™); Lane N for Negative control and lanes S1-S8 are for samples. Here, S1-S4 are negative lanes, lanes S5-S6 are positive for the *Burkholderia* genus & S7-S8 are positive for *Burkholderia cepacia*.

organism may also be transmitted during the milking process, and this is supported by the fact that the farm practice manual recommends milking. Immunocompromised people should refrain from consuming raw, unpasteurised milk, as it may serve as a potential source of infection from this organism [34].

In our phylogenetic analysis, BAU-MSR-AAM-02 (PQ657138), BAU-MSR-AAM-03 (PQ657139), BAU-MSR-AAM-04 (PQ657140), and BAU-MSR-AAM-05 (PQ657137) were the sequenced isolates from our study that showed 100% similarity with *Burkholderia* strains from various global sources, such as strains KT9066861.1 (*Burkholderia cepacia* strain PSTJ15) from India, LC420066 (*Burkholderia cepacia* strain JCM2801) from Japan, and MW386480.1 (*Burkholderia paraquae* strain NLG515) from China. These isolates clustered together in the phylogenetic tree, indicating that they were closely linked phylogenetically. Interestingly, all of the isolates from Bangladesh grouped together in a unique cluster within the tree, demonstrating a significant degree of genetic similarity between them. This could indicate that Bangladesh has a common evolutionary ancestry or environmental adaptation. With bootstrap support values ranging from 88% to 100%, the phylogenetic analysis also showed that these isolates had a strong phylogenetic link within the group and were genetically more connected to specific global *Burkholderia* strains.

Bcc has been identified in cattle, poultry and companion animals as well as in crustaceans and humans [4,35,21,23]. However, few studies exist on its prevalence in goats. This study's findings fill a significant knowledge gap and highlight the necessity of routine screening and biosecurity programmes in livestock populations to reduce this potential risk.

Despite our findings, several limitations should be acknowledged. The relatively small sample size and sampling from a single farm may limit the generalizability of the results. Additionally, as a cross-sectional study, this research cannot establish a direct causal relationship between Bcc infection and clinical outcomes such as abortion. Furthermore, environmental and human sampling were not included, and antimicrobial susceptibility testing could not be performed due to financial constraints. Future studies involving larger sample sizes, multiple locations, and One Health approaches are recommended.

## Conclusion

This study presents the first confirmation of *B. cepacia* complex (Bcc) in Bangladeshi goats. It was highly suspected that the environment of the farm may have been contaminated with *Burkholderia cepacia* complex; however, this was not directly investigated in the present study. The existence of Bcc in goats calls for more awareness and better biosecurity policies to reduce potential risks to human and animal health. In order to better understand and minimise the impact of Bcc in livestock production systems, future studies should concentrate on genomic characterization, environmental monitoring, and host-pathogen interactions.

## Supporting information

**S1 Fig. Culture of inoculum in Luria-Bertani broth.** Visible turbidity and distinct color transition in inoculated samples compared to the sterile control.
(TIF)

**S2 Fig. Representative results of the Glanders Rapid Detection Test showing a positive (+ve) band at the test line (T) indicating antigen detection.**
(TIF)

**S3 Fig. Indirect ELISA for the detection of Bcc-specific antibodies.** Microplate visualization following the addition of stop solution; yellow coloration indicates positive (+ve) reactivity, while clear wells represent negative (-ve) results.
(TIF)

**Acknowledgments:** We would like to extend our sincere gratitude to the farm owner and the workers who helped in this study and generously provided their time, resources, and support. Without their cooperation, this research would not have been possible.

We also wish to thank our dedicated labmates from the Department of Medicine for their invaluable support throughout the course of this study. Their assistance in laboratory procedures, data analysis, and their continuous encouragement were essential to the successful completion of this research.

## Author contributions

**Conceptualization:** Abdullah Al Mamun, Md. Mahbub Alam, Md. Siddiqur Rahman.

**Data curation:** Abdullah Al Mamun, Hasan Khan.

**Formal analysis:** Abdullah Al Mamun, Naeem Ahammed Ibrahim Fahim.

**Funding acquisition:** Md. Siddiqur Rahman.

**Investigation:** Abdullah Al Mamun, S. M. Sujan Ashraf, Amir Hamza Shuvo, Farhan Ibne Siddique.

**Methodology:** Abdullah Al Mamun, Amir Hamza Shuvo, Chandra Shaker Chouhan, Farzana Yeasmin.

**Software:** Abdullah Al Mamun, Naeem Ahammed Ibrahim Fahim.

**Supervision:** Md. Mahbub Alam, Md. Siddiqur Rahman.

**Writing – original draft:** Abdullah Al Mamun.

**Writing – review & editing:** Md. Mahbub Alam, Md Tanvir Rahman, Md. Siddiqur Rahman.

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
