## [Editor Report · Decision Letter 0]

7 Jul 2025

PONE-D-25-28883*Burkholderia cepacia* complex (Bcc) in Goats: First Report in BangladeshPLOS ONE

Dear Dr. Rahman,

Thank you for submitting your manuscript to PLOS ONE. After careful consideration, we feel that it has merit but does not fully meet PLOS ONE’s publication criteria as it currently stands. Therefore, we invite you to submit a revised version of the manuscript that addresses the points raised during the review process.

We look forward to receiving your revised manuscript.

Kind regards,

Rajesh Singh Rathore, Ph.D

Academic Editor

PLOS ONE

[This research was supported by the LDDP Research and Innovation Sub-project, Livestock and Dairy Development Project, Department of Livestock Services, Farmgate, Dhaka—Transmission dynamics of brucellosis and abortion risk factors in large dairy herds in Bangladesh (project code:RP-C-01-10).].

Additional Editor Comments:

1. Title Modification: There are several reports on Burkholderia cepacia complex (Bcc) in animals, including goats. Therefore, we recommend modifying the title of the paper to reflect this broader context.

2. Missing Table: Table 1 is not provided. Please ensure that all tables and figures are submitted as separate files as per the submission guidelines.

3. Figure 2: The marker in Figure 2 is unclear. Kindly provide a clearer gel image with labels indicating all the marker fragments.

4. Figure 4: Please label all marker fragments in Figure 4. Additionally, could you clarify why the gel images in Figures 2 and 4 have different backgrounds? Specifically, why does the DNA gel in Figure 4 have a blue background?

5. Control Group Gel Image: Please provide the gel image for the control group used in the detection of Burkholderia cepacia complex (Bcc).

6. Material and Methods References: Kindly provide the proper citations for all materials, methods, and kits used in this study, including the ladder.

7. PCR Protocol and Bioinformatics Reference: Please include the reference for the PCR protocol as well as the bioinformatics tools used in this study.

8. Gene Nomenclature: Please ensure that the writing style of the gene name is consistent and follows the standard conventions.

9. NCBI Submission: Please provide the accession number for the 16S gene data submission to NCBI.
---

## [Author Response · Author response to Decision Letter 1]

22 Jul 2025

Response to Editor Comments

Dear Dr. Rathore,

Thank you for your detailed and constructive feedback on our manuscript titled "Burkholderia cepacia complex (Bcc) in Goats: First Report in Bangladesh" (Manuscript ID: PONE-D-25-28883R1). We appreciate the time and effort you and the reviewers have invested in evaluating our work. Below, we provide our responses to each of the comments raised during the review process:

Reply: Thank you for pointing this out. We have reviewed the manuscript formatting and ensured that it now fully complies with PLOS ONE's style requirements, including those for file naming. We have adhered to the specified guidelines and made sure all files are correctly labeled as per the instructions provided.

2. Thank you for stating the financial disclosure. Please also clarify the role of the funders in the study.

Reply: We have updated the financial disclosure to reflect the accurate funding source. This research was supported by the University Grants Commission of Bangladesh (UGC) under the project titled "Prevalence and Risk Factors of Glanders in Humans and Horses" (Project Code: 2024/17/UGC) and Ministry of Education (MoE) under the project titled "Development of diagnonis and control strategies against brucellosis in small ruminants and humans" (project code:2022/16/MoE). As per your request, we have added the statement that the funders had no role in the study design, data collection and analysis, decision to publish, or preparation of the manuscript. This updated disclosure has been included in the revised manuscript.

3. Title Modification: There are several reports on Burkholderia cepacia complex (Bcc) in animals, including goats. Therefore, we recommend modifying the title of the paper to reflect this broader context.

Reply: Thank you for your valuable feedback. As far as I am aware, this is the first report of Burkholderia cepacia complex (Bcc) detection in animals, specifically goats, in Bangladesh. While there may be several reports of Bcc in animals worldwide, this study presents the first detection of Bcc in goats within the context of Bangladesh.

Therefore, I would like to retain the focus of the title to reflect this novel finding in Bangladesh. However, based on your suggestion, I have revised the title to better clarify the unique contribution of this study.

Thank you once again for your insightful comments.

4. Missing Table: Table 1 is not provided. Please ensure that all tables and figures are submitted as separate files as per the submission guidelines.

Reply: Thank you for your observation. I apologize for the oversight and have now included Table 1 as a separate file (file name- Table) under the section other, while also ensuring it is placed in the manuscript after the paragraph in which it is first cited, as per the guidelines.

5. Figure 2: The marker in Figure 2 is unclear. Kindly provide a clearer gel image with labels indicating all the marker fragments

Reply: We are thankful for the editor’s constructive comment about for clarification regarding Figure 2. I apologize for the earlier lack of clarity. I have now provided a clearer gel image with labels indicating all the marker fragments, as per your suggestion.

6. Figure 4: Please label all marker fragments in Figure 4. Additionally, could you clarify why the gel images in Figures 2 and 4 have different backgrounds? Specifically, why does the DNA gel in Figure 4 have a blue background?

Reply: We appreciate the editor’s attention to the issue of the Figure 4. I have now labeled all the marker fragments in the figure as requested. Additionally, regarding the background color difference between Figures 2 and 4, I have corrected the issue. The gel image in Figure 4 now has a consistent background, and I have ensured that both figures align in terms of presentation.

7. Control Group Gel Image: Please provide the gel image for the control group used in the detection of Burkholderia cepacia complex (Bcc).

Reply: Thank you for your comment. I have included the gel image for the control group in Figure 2, where both the suspected animals and the control animals have been tested. I hope this clarifies the inclusion of the control group.

8. Material and Methods References: Kindly provide the proper citations for all materials, methods, and kits used in this study, including the ladder.

Reply: Thank you for your valuable feedback. I have now included the proper citations for all materials, methods, and kits used in the study in line no 167, 189, including the ladder in line 192 (https://www.thermofisher.com/order/catalog/product/15628019), as requested. The references have been added to the Materials and Methods section.

9. PCR Protocol and Bioinformatics Reference: Please include the reference for the PCR protocol as well as the bioinformatics tools used in this study.

Reply: Thank you for your feedback. I have now included the reference for the PCR protocol, as well as the bioinformatics tools used in this study, in the revised manuscript. These references have been added to the appropriate sections for clarity.

10. Gene Nomenclature: Please ensure that the writing style of the gene name is consistent and follows the standard conventions

Reply: Thank you for your valuable feedback. I have now ensured that the writing style of the gene names is consistent throughout the manuscript and follows the standard conventions.

11. NCBI Submission: Please provide the accession number for the 16S gene data submission to NCBI.

Reply: Thank you for your comment. The sequences have been deposited in GenBank under the following accession numbers:

• PQ657137.1 (Burkholderia contaminans BAU-MSR-AAM-01)

• PQ657138.1 (B. ambifaria BAU-MSR-AAM-02)

• PQ657139.1 (B. contaminans BAU-MSR-AAM-03)

• PQ657140.1 (B. paraquae BAU-MSR-AAM-04)

• PQ764482.1 (B. cepacia BAU-MSR-AAM-05)

We trust that the revisions made in response to the editor’s and reviewers’ comments have addressed all concerns and enhanced the clarity and quality of the manuscript. We appreciate your thoughtful feedback and look forward to your further guidance.

Kind regards,

Dr. Md. Siddiqur Rahman

Professor, Department of Medicine

Bangladesh Agricultural University, Mymensingh, Bangladesh

Email: siddique.medicine@bau.edu.bd

Contact: +8801918181550

---

## [Decision Letter · Decision Letter 1]

19 Nov 2025

PONE-D-25-28883R1*Burkholderia cepacia* complex (Bcc) in Goats: First Report in BangladeshPLOS ONE

Dear Dr. Rahman,

Thank you for submitting your manuscript to PLOS ONE. After careful consideration, we feel that it has merit but does not fully meet PLOS ONE’s publication criteria as it currently stands. Therefore, we invite you to submit a revised version of the manuscript that addresses the points raised during the review process.

Please review the reviewer comments below and attached and revise your manuscript to address the questions raised by Reviewer 2, providing a point-by-point response to the reviewer upon resubmission.  Please submit your revised manuscript by Jan 03 2026 11:59PM. If you will need more time than this to complete your revisions, please reply to this message or contact the journal office at plosone@plos.org. Please include the following items when submitting your revised manuscript:

We look forward to receiving your revised manuscript.

Kind regards,

Sarah Jose, Ph.D.

Staff Editor

PLOS ONE

Journal Requirements:

Reviewers' comments:

Reviewer's Responses to Questions

**Comments to the Author**

1. If the authors have adequately addressed your comments raised in a previous round of review and you feel that this manuscript is now acceptable for publication, you may indicate that here to bypass the “Comments to the Author” section, enter your conflict of interest statement in the “Confidential to Editor” section, and submit your "Accept" recommendation.

Reviewer #1: All comments have been addressed

Reviewer #2: All comments have been addressed

2. Is the manuscript technically sound, and do the data support the conclusions?

Reviewer #1: Yes

Reviewer #2: Partly

3. Has the statistical analysis been performed appropriately and rigorously? 

Reviewer #1: Yes

Reviewer #2: Yes

4. Have the authors made all data underlying the findings in their manuscript fully available?

Reviewer #1: Yes

Reviewer #2: Yes

5. Is the manuscript presented in an intelligible fashion and written in standard English?

Reviewer #1: Yes

Reviewer #2: No

6. Review Comments to the Author

Reviewer #1: (No Response)

Reviewer #2: my comments:

1. The manuscript requires improvement in English language quality.

2. Why the authors use Rapid Glanders detection test for B. cepacia complex. Please clarify.

3.Please clarify the criteria used for selecting the animals. Please explain how the authors ensured that other infectious agents were ruled out and describe any clinical or epidemiological criteria used to include animals in the study

4. PCR is generally more sensitive than serological assays; however, in this study the PCR positivity rate is lower.

5. Please explain why whole blood was not used for culture and why only serum was selected.

6. In the phylogenetic analysis, please specify the outgroup used.

7. PLOS authors have the option to publish the peer review history of their article (what does this mean?). If published, this will include your full peer review and any attached files.

Reviewer #1: No

Reviewer #2: **Yes:** Abdollah Derakhshandeh

---

## [Author Response · Author response to Decision Letter 2]

24 Dec 2025

We sincerely thank the Academic Editor and Reviewer 2 for their careful evaluation of our manuscript and for the constructive comments provided.

All comments raised by Reviewer 2 have been addressed thoroughly and point by point. Specifically:

The manuscript has undergone comprehensive English language editing to improve clarity, grammar, and scientific readability throughout.

We have clarified the rationale for using the Rapid Glanders Detection Test as an initial screening assay only, emphasizing that all positive findings were subsequently confirmed using ELISA, genus- and species-specific PCR, Sanger sequencing, and phylogenetic analysis.

Detailed clinical and epidemiological criteria for animal selection have been clearly described, and study limitations regarding exclusion of other infectious agents have been transparently acknowledged.

The discrepancy between serological and PCR positivity rates has been explained based on biological and diagnostic principles and clarified in the Discussion section.

The justification for using serum rather than whole blood for culture and molecular analysis has been explicitly stated.

The outgroup used in the phylogenetic analysis has now been clearly specified.

All revisions have been incorporated into the revised manuscript, highlighted in the tracked-changes version, and reflected in the clean version submitted. We believe these changes have significantly strengthened the manuscript.

We appreciate the opportunity to revise our work and respectfully submit the revised manuscript for further consideration.

Sincerely,

Dr. Md. Siddiqur Rahman

---

## [Decision Letter · Decision Letter 2]

4 Mar 2026

PONE-D-25-28883R2*Burkholderia cepacia* complex (Bcc) in Goats: First Report in BangladeshPLOS One

Dear Dr. Rahman,

Thank you for submitting your manuscript to PLOS ONE. After careful consideration, we feel that it has merit but does not fully meet PLOS ONE’s publication criteria as it currently stands. Therefore, we invite you to submit a revised version of the manuscript that addresses the points raised during the review process.

We look forward to receiving your revised manuscript.

Kind regards,

Marcos Pileggi, Ph.D

Academic Editor

PLOS One

Journal Requirements:

Reviewers' comments:

Reviewer's Responses to Questions

**Comments to the Author**

1. If the authors have adequately addressed your comments raised in a previous round of review and you feel that this manuscript is now acceptable for publication, you may indicate that here to bypass the “Comments to the Author” section, enter your conflict of interest statement in the “Confidential to Editor” section, and submit your "Accept" recommendation.

Reviewer #1: All comments have been addressed

2. Is the manuscript technically sound, and do the data support the conclusions?

Reviewer #1: Yes

3. Has the statistical analysis been performed appropriately and rigorously? 

Reviewer #1: Yes

4. Have the authors made all data underlying the findings in their manuscript fully available?

Reviewer #1: Yes

5. Is the manuscript presented in an intelligible fashion and written in standard English?

Reviewer #1: Yes

6. Review Comments to the Author

Reviewer #1: (No Response)

7. PLOS authors have the option to publish the peer review history of their article (what does this mean?). If published, this will include your full peer review and any attached files.

Reviewer #1: No

---

## [Author Response · Author response to Decision Letter 3]

22 Mar 2026

All reviewer comments have been carefully addressed. A detailed point-by-point response has been provided in the uploaded “Response to Reviewers” document, and corresponding changes have been incorporated into the revised manuscript.

---

## [Decision Letter · Decision Letter 3]

21 Apr 2026

*Burkholderia cepacia* complex (Bcc) in Goats: First Report in Bangladesh

PONE-D-25-28883R3

Dear Dr. Rahman,

We’re pleased to inform you that your manuscript has been judged scientifically suitable for publication and will be formally accepted for publication once it meets all outstanding technical requirements.

Kind regards,

Marcos Pileggi, Ph.D

Academic Editor

PLOS One

Additional Editor Comments (optional):

Reviewers' comments:

Reviewer's Responses to Questions

**Comments to the Author**

1. If the authors have adequately addressed your comments raised in a previous round of review and you feel that this manuscript is now acceptable for publication, you may indicate that here to bypass the “Comments to the Author” section, enter your conflict of interest statement in the “Confidential to Editor” section, and submit your "Accept" recommendation.

Reviewer #1: All comments have been addressed

2. Is the manuscript technically sound, and do the data support the conclusions?

Reviewer #1: Yes

3. Has the statistical analysis been performed appropriately and rigorously? 

Reviewer #1: Yes

4. Have the authors made all data underlying the findings in their manuscript fully available?

Reviewer #1: Yes

5. Is the manuscript presented in an intelligible fashion and written in standard English?

Reviewer #1: Yes

6. Review Comments to the Author

Reviewer #1: (No Response)

7. PLOS authors have the option to publish the peer review history of their article (what does this mean?). If published, this will include your full peer review and any attached files.

Reviewer #1: No

---

## [Editor Report · Acceptance letter]

PONE-D-25-28883R3

PLOS One

Dear Dr. Rahman,

I'm pleased to inform you that your manuscript has been deemed suitable for publication in PLOS One. Congratulations! Your manuscript is now being handed over to our production team.

Kind regards,

on behalf of

Dr. Marcos Pileggi

Academic Editor

PLOS One